# Human Macrophages Activate Bystander Neutrophils’ Metabolism and Effector Functions When Challenged with *Mycobacterium tuberculosis*

**DOI:** 10.3390/ijms25052898

**Published:** 2024-03-01

**Authors:** Dearbhla M. Murphy, Anastasija Walsh, Laura Stein, Andreea Petrasca, Donal J. Cox, Kevin Brown, Emily Duffin, Gráinne Jameson, Sarah A. Connolly, Fiona O’Connell, Jacintha O’Sullivan, Sharee A. Basdeo, Joseph Keane, James J. Phelan

**Affiliations:** 1Department of Clinical Medicine, School of Medicine, Trinity Translational Medicine Institute (TTMI), Trinity Centre for Health Sciences, St. James’s Hospital, Trinity College Dublin, The University of Dublin, Dublin 8, D08 W9RT Dublin, Ireland; murphd58@tcd.ie (D.M.M.); walsha55@tcd.ie (A.W.); steinla@tcd.ie (L.S.); docox@tcd.ie (D.J.C.); kebrown@tcd.ie (K.B.); duffine@tcd.ie (E.D.); jamesog@tcd.ie (G.J.); sconnol4@tcd.ie (S.A.C.); josephmk@tcd.ie (J.K.); 2School of Biochemistry and Immunology, Trinity Biomedical Sciences Institute (TBSI), Trinity College Dublin, The University of Dublin, D02 R590 Dublin, Ireland; petrasan@tcd.ie; 3Department of Surgery, Trinity St. James’s Cancer Institute, Trinity Translational Medicine Institute (TTMI), St. James’s Hospital, Dublin 8, D08 W9RT Dublin, Ireland; oconnefi@tcd.ie (F.O.); osullij4@tcd.ie (J.O.)

**Keywords:** immunometabolism, glycolysis, neutrophil priming and activation, neutrophil function, neutrophil metabolism, granulocytes, infection, tuberculosis, polymorphonuclear cells, *Mycobacterium tuberculosis*

## Abstract

Neutrophils are dynamic cells, playing a critical role in pathogen clearance; however, neutrophil infiltration into the tissue can act as a double-edged sword. They are one of the primary sources of excessive inflammation during infection, which has been observed in many infectious diseases including pneumonia and active tuberculosis (TB). Neutrophil function is influenced by interactions with other immune cells within the inflammatory lung milieu; however, how these interactions affect neutrophil function is unclear. Our study examined the macrophage–neutrophil axis by assessing the effects of conditioned medium (MΦ-CM) from primary human monocyte-derived macrophages (hMDMs) stimulated with LPS or a whole bacterium (*Mycobacterium tuberculosis*) on neutrophil function. Stimulated hMDM-derived MΦ-CM boosts neutrophil activation, heightening oxidative and glycolytic metabolism, but diminishes migratory potential. These neutrophils exhibit increased ROS production, elevated NET formation, and heightened CXCL8, IL-13, and IL-6 compared to untreated or unstimulated hMDM-treated neutrophils. Collectively, these data show that MΦ-CM from stimulated hMDMs activates neutrophils, bolsters their energetic profile, increase effector and inflammatory functions, and sequester them at sites of infection by decreasing their migratory capacity. These data may aid in the design of novel immunotherapies for severe pneumonia, active tuberculosis and other diseases driven by pathological inflammation mediated by the macrophage–neutrophil axis.

## 1. Introduction

Lung infections are a major cause of morbidity and mortality. Neutrophils are the first cell recruited to a tissue during infection, allowing them to rapidly respond to the invading pathogen [1]. They can rapidly respond to infection in tissues such as the lung, using a diverse arsenal of effector functions such as phagocytosis, oxidative burst, pathogen entrapment within neutrophil extracellular traps (NETs), and cytokine production. Despite their abundance, a significant research gap exists around characterizing their function in a range of disease states. Validating these findings across disease models and examining how immune cells communicate and specifically interact with neutrophils needs to be examined.

Pneumonia is an inflammation of the lungs which is usually caused by Gram-negative bacteria, such as *Klebsiella pneumoniae*, *Legionella pneumophila*, and *Haemophilus influenzae*. Mycobacteria such as *Mycobacterium tuberculosis* (Mtb), the causative agent of tuberculosis (TB), can also cause pneumonia during acute infection. Chronic TB disease results in extensive pathological inflammation in the lung. During pneumonia and TB disease, airway macrophages, the first cells to interact with invading pathogens in the lung, readily recruit neutrophils [2]. Neutrophils are one of the first cells recruited to sites of lung infection where they rapidly phagocytose bacteria [3]. However, the role of neutrophils in pathology versus protection remains uncertain and contentious. While they have been shown to play a protective role during early Mtb [4], in the later stages of TB disease, neutrophils are key drivers of pathological inflammation which leads to damage to the host lung and poorer disease outcomes [5]. As the disease progresses, cytokine and chemokine production from both macrophages and neutrophils causes further recruitment of both cell types, which can further drive the inflammatory response [6,7]. Neutrophils can act as a double-edged sword, where they are protective in the early stages of infection and then drive pathological inflammation causing severe pneumonia [2].

While it is known that neutrophils can recruit monocytes to the site of infection, and these monocytes can differentiate into macrophages and recruit further neutrophils [8], the direct effect of infected human macrophages on bystander neutrophils’ activation and phenotype remains elusive. Addressing this important knowledge gap will allow us to understand how recruited airway macrophages propagate inflammation resulting in pathological lung damage. To this end, using an ex vivo model of primary human monocyte-derived macrophages (hMDMs) and neutrophils, the current study explores the macrophage–neutrophil axis during infection and assesses if the macrophage-conditioned medium (MΦ-CM) from Mtb- or LPS-stimulated macrophages exhibits any modulatory effects on conventional neutrophil function.

## 2. Results

The host response to infections is complex, involving the release of various secreted factors from a wide variety of cells, including neutrophils, macrophages, monocytes, T cells, B cells, NK cells, and dendritic cells. Resident macrophages provide the first line of defence against invading pathogens in the lung, for example, where they recruit neutrophils and monocytes to the site of infection. The macrophage–neutrophils axis is undefined and relatively unexplored in the context of human lung infections to date. The current study set out to model the human macrophage–neutrophil axis in vitro (Figure 1A,B).

### 2.1. Mtb- and LPS-Stimulated hMDMs Secrete Cytokines and Chemokines Known to Modulate Neutrophil Biology

Prior to examining the effect of the MΦ-CM on neutrophil function, we first needed to determine if the MΦ-CM had the capacity to alter neutrophil biology by characterizing and quantifying secreted levels of a variety of cytokines and chemokines in the MΦ-CM in response to Mtb. LPS is a crucial stimulant of immune responses in many cell types. LPS was therefore used as a second stimulant in our neutrophil model as it mimics bacterial infection, allowing us to model a Gram-negative TLR4-mediated bacterial response which is a common cause of pneumonia. To accomplish this, we repurposed data from a previous published study from our laboratory [9]. hMDMs, differentiated from fresh PBMCs, were isolated from healthy blood donors over 7 days, and were stimulated with Mtb or LPS. Then, 24 h post-stimulation, the MΦ-CM was processed and the levels of 18 different cytokines and chemokines in 18 independent donors were quantified using MSD MultiArray Technology (Figure 2). We found that concentrations of TNF-α, IL-1β, IL-10, IL-4, IL-2, CXCL8, IL-13, IFN-γ and IL-6 were significantly elevated in MtbCoM and LPSCoM compared with the unstimulated controls (Figure 2A). Furthermore, the concentrations of chemokines IP-10, MCP-4, TARC, MDC, Eotaxin, Eotaxin-3, MIP-1β, and MCP-1 were significantly increased in MtbCoM and LPSCoM compared with the unstimulated controls (Figure 2B). Although elevated, the levels of MIP-1α were not significantly increased in response to Mtb or LPS stimulation (Figure 2B).

### 2.2. MΦ-CM from Mtb- and LPS-Stimulated hMDMs Activates Neutrophils and Alters Their Migratory Potential

Having established in hMDMs that Mtb or LPS can induce the expression of various cytokines and chemokines, many with known effects on neutrophil biology, we examined if this MΦ-CM had the capacity to activate neutrophils. To undertake these and other experiments for the remainder of the study, we examined the effect of four MΦ-CM types on various parameters of neutrophil function; 20% RPMI (cRPMI), 20% MΦ-CM from unstimulated hMDMs (UsCoM), 20% MΦ-CM from Mtb-stimulated hMDMs (MtbCoM), and 20% MΦ-CM from LPS-stimulated hMDMs (LPSCoM) (the remaining 80% is RPMI supplemented with 1% FBS). The cRPMI treatment was also supplemented with 10% human serum to control for the 10% human serum in the UsCoM, MtbCoM, and LPSCoM treatments. The MΦ-CM was incubated with freshly isolated neutrophils one hour prior to experimental analysis. Upon exposing neutrophils for one hour to MΦ-CM, we observed that neutrophils exposed to MtbCoM or LPSCoM exhibited profound phenotypic differences in cell morphology compared to neutrophils exposed to cRPMI and UsCoM controls (Figure 3A). Neutrophils appeared more elongated and displayed some evidence of neutrophil extracellular traps (NETs), all surrogate indicators of activation (Figure 3A) [10]. To examine if MtbCoM or LPSCoM had the potential to activate neutrophils, we assessed the expression of the granulation and degranulation markers, CD62L and CD63, respectively, using flow cytometric analyses. Briefly, following doublet exclusion, neutrophils were selected based on size and granularity on a plot of forward scatter (FSC) versus side scatter (SSC) (Figure 3B). Subsequently, dead cells were excluded using a viability dye and neutrophils were defined as CD15+ cells (Figure 3B). We found that MtbCoM or LPSCoM treatment significantly reduced CD62L expression (Figure 3C) while simultaneously promoting the expression of CD63 (Figure 3D) in neutrophils, thereby confirming that MtbCoM or LPSCoM can activate neutrophils. Using cell culture transwell assay inserts, we found that MtbCoM- and LPSCoM-treated neutrophils exhibited a significantly reduced capacity to migrate toward fMLP compared to neutrophils treated with UsCoM (Figure 3E). Further flow cytometric analysis showed that this reduced capacity to migrate could be due to the downregulation of the CXCR2 receptor, a key mediator of migration in neutrophils (Figure 3F) [11].

### 2.3. MΦ-CM from Mtb- and LPS-Stimulated hMDMs Augments Neutrophil Metabolism in Unstimulated and Stimulated Neutrophils

Being highly active and dynamic cells, neutrophils utilise a diverse array of metabolic pathways for growth, proliferation, survival and death [12]. Accordingly, these metabolic pathways are fundamental to neutrophil effector functions during stimulation, activation, and infection [13]. We found that MtbCoM or LPSCoM significantly increased oxidative phosphorylation and glycolysis in neutrophils, as determined by measuring OCR (Figure 4A) and ECAR (Figure 4B), respectively. These energetic shifts are illustrated in metabolic phenograms that show MtbCoM and LPSCoM significantly augment neutrophil glycolytic and oxidative metabolism (Figure 4C).

We have shown that MtbCoM or LPSCoM can augment metabolism in unstimulated neutrophils. However, metabolic rewiring can also occur upon activation and infection in neutrophils, to adapt to the invading pathogen and produce enough energy to fulfil its host-defensive duties [14]. Accordingly, we examined if MtbCoM- or LPSCoM-treated neutrophils exhibited altered immunometabolic profiles upon stimulation and activation (Figure 5). To do this, we stimulated neutrophils with Mtb, to mimic Mtb infection in the human lung (Figure 5A). Prior to assessing if MtbCoM or LPSCoM affects the ECAR and OCR profiles, we first established a baseline and examined if Mtb stimulation alone affected neutrophil metabolism. We found that soon after Mtb stimulation, Mtb significantly increased both the ECAR and OCR profiles in neutrophils (Figure 5B), indicative of an increase in OXPHOS and glycolysis in these cells. Interestingly, we found no significant effect on OCR and ECAR in neutrophils pre-treated with cRPMI, UsCoM, MtbCoM or LPSCoM upon Mtb stimulation (Figure 5C), indicating that Mtb stimulation alone augmented and potentially maxed out the immunometabolic response in these cells, regardless of MΦ-CM pre-treatment.

Next, we stimulated MtbCoM- or LPSCoM-treated neutrophils with fMLP, a G-protein receptor agonist, known to rapidly activate neutrophils. Neutrophil activation by fMLP is dependent on store-operated calcium influx that appears to be regulated by Cl− channels and linked, in part, to non-selective cation channels [15]. We found post-treatment that that the highest metabolic response to fMLP was 40 min post-stimulation, observed by a significant increase in OCR in neutrophils pre-treated with MtbCoM (Figure 5D). No change in ECAR was observed in neutrophils pre-treated with LPSCoM (Figure 5E). Overall, these results show that MtbCoM and LPSCoM can augment neutrophil metabolism and could differently affect oxidative metabolism during activation depending on the stimulus.

### 2.4. MtbCoM or LPSCoM Modulates Neutrophil Cytokine Secretion, NETosis, and ROS Production in Neutrophils

Upon confirming that MtbCoM or LPSCoM can activate neutrophils, modulate their immunometabolic potential, and their migratory ability, we further examined whether the MΦ-CM can alter additional key effector functions of neutrophils. First, we examined if MΦ-CM-treated cells exhibited differences in their ability to secrete cytokines (Figure 6). To do this, neutrophils were incubated for one hour with cRPMI, UsCoM, MtbCoM, or LPSCoM, the MΦ-CM was removed, and the cells washed twice. The cells were incubated for four hours, the supernatant collected, and the levels of the cytokines CXCL8, IL-13, IL-6, TNF-α, IL-1β and IL-10 were quantified (Figure 6A). We observed that MtbCoM- or LPSCoM-treated neutrophils secreted significantly increased concentrations of CXCL8, IL-13 and IL-6 compared with controls (Figure 6A).

Having shown that MtbCoM or LPSCoM could induce cytokine secretion in unstimulated neutrophils, we hypothesized that we would see similar levels in neutrophils that were also activated with fMLP, as MtbCoM and LPSCoM are capable of activating neutrophils alone. To examine this, we repeated the procedure but stimulated the neutrophils at the beginning of their four-hour incubation with fMLP. fMLP increased CXCL8 and IL-13 levels in cRPMI-treated control neutrophils, as expected, but fMLP could not boost CXCL8, IL-13, or IL-6 levels in cells treated with MtbCoM or LPSCoM (Figure 6B). Given these findings, it is evident that MtbCoM and LPSCoM can induce cytokine levels reminiscent of conventional neutrophil activation.

To explore the effect of MtbCoM or LPSCoM on neutrophil effector functions further, flow cytometric analyses were carried out to examine if MΦ-CM-treated neutrophils exhibited alterations in various cellular parameters including NETosis, ROS production, glucose uptake and apoptotic activity (Figure 7). We found that MtbCoM- or LPSCoM-treated cells exhibited significantly higher levels of NETosis, confirmed by cells testing double positive for the dyes Sytox red and DAPI (Figure 7A). Furthermore, we showed that MtbCoM or LPSCoM promoted ROS production in these neutrophils, indicated by significantly increased median fluorescence intensity (MFI) for DHR-123 (Figure 7B). Neutrophils treated with cRPMI, UsCoM, MtbCoM and LPSCoM exhibited comparable levels of glucose uptake (Figure 7C). Finally, as cell death can indirectly and directly contribute to some of the findings presented here (activation status, metabolism ROS production, for example), we also examined if the MΦ-CM could alter early and late apoptosis but found no significant changes across all treatment groups (Figure 7D). All raw data used in this manuscript can be accessed in the Appendix A.

## 3. Discussion

Neutrophils are at the forefront of host defense but can contribute to pathology and mortality by causing tissue damage during severe pneumonia [4]. Little is known about human neutrophil metabolism and function and how neutrophils are regulated by other early-responding immune cells in their environment, such as the macrophage. Understanding how innate immune cells propagate inflammation is fundamental to determine therapeutically targetable pathways aimed at blocking excessive neutrophil-mediated inflammation. This study hypothesized that stimulated macrophages propagate inflammation by activating neutrophils. We established a model whereby primary human neutrophils were treated with the MΦ-CM from resting or stimulated macrophages.

We hypothesized that the MΦ-CM from stimulated hMDMs had the potential to exhibit profound effects on bystander neutrophils and alter their effector function [12]. Many of the cytokines we found present in our MΦ-CM from stimulated hMDMs are known to prime and activate neutrophils [16,17,18,19]. Notably, we also observed a high level of expression of IL-6 in both Mtb- and LPSCoM. IL-6 is a key cytokine in neutrophil recruitment and trafficking [20,21]. Its elevated expression in both groups suggests a potent pro-inflammatory environment conducive to enhanced neutrophil recruitment and activation which can exacerbate inflammation in the lung. This is supported by the findings of Mateer et al. who suggest that in an in vivo model of the pulmonary manifestation of IBD, IL-6 induced the recruitment of neutrophils to the lung, with exacerbated recruitment leading to neutrophilia and bacteremia [22]. Similar findings by Wright et al. noted that IL-6 is associated with neutrophilia under inflammation [23]. Additionally, there is evidence to suggest that cytokines such as TNF-α, CXCL8, and IFN-γ present in MtbCoM or LPSCoM may mechanistically mediate neutrophil activation. In our model, however, further studies using blocking/neutralizing antibodies are warranted to further understand the underlying mechanism(s) [24,25,26,27,28]. Despite many studies describing IL-1β as a potent activator of neutrophils and inducer of NETosis [18,29], the opposite has also been shown [30]. Moreover, although IL-10 and IL-4 are thought to have a negligible effect on neutrophil activation, evidence from other models suggests that IL-2, IL-13, and MCP-1 could play a role in priming and activating neutrophils in the current model [31,32]. Although outside the scope of the current study, further functional and mechanistic studies are warranted to fully characterize the effects of these secreted factors on neutrophil function and to what extent, if any, how these secreted factors affect neutrophil degranulation/granulation specifically.

In Figure 3C, we demonstrate the loss of the granulation marker CD62L. Notably, CD62L loss is crucial for neutrophil adhesion and migration [33]. In bronchoalveolar lavage fluid from smokers, non-smokers, and the peripheral blood of healthy volunteers, human neutrophils expressing low CD62L levels switch to an activated phenotype after transmigration to the lung irrespective of inflammatory disease [34]. The same result was observed in neutrophils from sarcoidosis patients, where lung neutrophils displayed an activated phenotype in both homeostatic and inflammatory conditions [34]. We also observed a profound reduction in neutrophil migratory capacity upon MtbCoM or LPSCoM treatment. Published research has shown that neutrophils do not only exit the vasculature into tissues, but they can also re-enter the vasculature after surveying tissue in a process called ‘reverse migration’ [35]. Our data indicates that neutrophils exposed to an inflammatory microenvironment lost their ability to undergo reverse migration out of the lung in response to Mtb infection suggesting that they may be retained in the tissue where they can propagate pathogenic inflammation. The findings regarding the loss of CXCR2, known to play a crucial role in facilitating neutrophil migration to sites of inflammation, such as the human lung [36,37], is also significant. Neutrophils identified in human lung alveoli typically express high CXCR2 levels [36]. CXCL8 binds to CXCR2 with high affinity, and we showed that CXCL8 levels were elevated in MtbCoM or LPSCoM. The downregulation observed as a response to MΦ-CM from stimulated macrophages may explain their reduced capacity to migrate. This indicates that CXCL8 and its associated receptors such as CXCR2 could play an important role in retaining neutrophils at sites of inflammation. Other than CXCL8, CXCL1, also known as GRO-α, also interacts with the CXCR2 receptor. CXCL1 is known to induce neutrophil influx during bacterial infections [38]. In addition to its role in directly recruiting neutrophils, it has been demonstrated that in cancer, CXCL1 also regulates the expression of CXCR2 on the surface of neutrophils. Studies have shown that CXCL1 can upregulate the expression of CXCR2, thereby increasing the sensitivity of neutrophils to CXCL1 and other CXCR2 ligands [39,40]. The levels of CXCL1 in our CoMs were not examined in this study; however, they could be an intriguing avenue for future of investigation. Exploring the impact of migration toward Mtb- and LPSCoM compared to UsCoM could be another compelling pathway for future research in our model.

Activated neutrophils elicit their effector functions through multiple metabolic pathways, including glycolysis, OXPHOS, glycogenesis, fatty acid metabolism, gluconeogenesis, and the pentose phosphate pathway [12,41]. Moreover, neutrophils are subjected to nutrient fluctuations in their immediate microenvironment, requiring them to differentially rewire their metabolic pathways quickly to adapt to new environmental demands [42]. Upon the examination of glycolysis (measured via ECAR) and OXPHOS (measured by OCR), we found that MtbCoM or LPSCoM enhanced both glycolytic and oxidative metabolism in unstimulated neutrophils, indicative of a typical immunometabolic response to activation [12]. Considering these findings, however, it is important to note that while ECAR is synonymous with glycolysis, it does not always correspond to glycolysis and may also indicate activation of the pentose phosphate pathway (PPP) or amino acid metabolism.

Moreover, we showed for the first time, that neutrophils exhibit augmented glycolytic and OXPHOS profiles when directly stimulated with Mtb. Pre-treatment with MΦ-CM, however, did not affect this metabolic response to Mtb stimulation. These results showed that although MtbCoM or LPSCoM boosted the unstimulated cell metabolic profile, subsequent infection with Mtb seems to override any effect of MΦ-CM pre-treatment, pushing the neutrophils to their maximal glycolytic and oxidative capacities regardless. Interestingly, when MΦ-CM pre-treated cells were stimulated with fMLP, a neutrophil activator, neutrophils pre-treated with MtbCoM showed significantly higher oxygen consumption upon activation, likely plateauing shortly thereafter due to the oxidative burst, which is a common feature in activated neutrophils. The sustained increase in metabolic activity observed with Mtb treatment is likely due to the additional cellular processes being utilized, for example, the phagocytosis of Mtb mycobacteria, the intracellular presence of Mtb and its containment within the neutrophil, and the multiple PAMPs and DAMPs activated on and within the neutrophil itself. Nevertheless, these results show that MtbCoM and LPSCoM enhance bioenergetics in unstimulated neutrophils. The results also indicate that depending on the nature of the pathogen (Mtb or fMLP), the neutrophil exhibits metabolic plasticity and seems to adapt to the nature of the threat, either through a sustained metabolic response or through transient oxidative bursts. Either way, fully characterizing the immunometabolic responses of neutrophils in infected human lung models remains in its infancy and should be explored in greater detail in future studies [41].

Finally, we demonstrate that MtbCoM or LPSCoM induced the expression of CXCL8, IL-13, and IL-6 from neutrophils, which are cytokines all known to contribute to an exacerbation of human lung pathologies, including TB and ARDS, which can result from chronic pneumonia [43,44,45,46]. Most surprisingly, LPSCoM induced IL-6 to a greater extent than MtbCoM. It is notable that the specific role of IL-6 produced by neutrophils is not extensively studied, leading to conflicting findings regarding its functional significance. The production of IL-6 by neutrophils appears to vary depending on the stimulating factors present. Research has documented the secretion of IL-6 by neutrophils following exposure to LPS and GM-CSF, as well as in response to stimulation of TLR8 by viral agents [47,48,49]. This responsiveness underscores the ability of neutrophils to perceive and respond to their microenvironment, adjusting their cytokine release patterns accordingly.

Furthermore, MtbCoM or LPSCoM induced high ROS production in neutrophils and caused the neutrophils to undergo heightened NETosis. Unsurprisingly, glycolysis underpins NET formation, a process also known to be very sensitive to fluctuations in glucose availability [50]. Moreover, ROS is also known to play a key role in NET release [51]. The mitochondrial network, which encompasses OXPHOS, also reinforces various neutrophil effector functions including chemotaxis, phagocytosis, the respiratory burst, and apoptosis [52]. Indeed, it is highly likely that the increase in oxygen consumption we observed is attributed to increased ROS production in these cells, and in neutrophils stimulated with fMLP after MtbCoM pre-treatment. Despite enhanced activation, immunometabolism, and effector functions in response to MtbCoM or LPSCoM treatment, we observed no increase in glucose uptake as a result of these treatments. Neutrophils respond and carry out their functions quickly in vivo which may be due to priming by the inflammatory environment. A recent seminal paper comparing neutrophil function and metabolism in acute and chronic inflammation also supports this hypothesis. They showed that neutrophils undergo gluconeogenesis and glycogenesis in response to acute physiological stress and activation, metabolic shifts that they showed to be required for optimal energy production, function, and survival [13]. Additionally, their findings indicate that chronic inflammation dysregulates these processes which have profound inhibitory effects on neutrophil function and survival [13]. Our findings align with this study, underscoring the importance of metabolic shifts in optimizing neutrophil function and survival in the early stages of infection. Our data further highlights and emphasizes the potency with which MtbCoM or LPSCoM can activate neutrophils and shows that an initial stimulatory or activation event could potentially lead to a downstream cascade of activated neutrophils, all of which may not have been exposed to the initial triggering event. Additionally, it is possible that the varied results we observed, such as the induction of OCR in MtbCoM-treated neutrophils (LPSCoM failed to induce OCR) could be the result of differential cytokine and chemokine levels in Mtb- vs. LPSCoM. For example, LPSCoM induced 2–4 times higher levels of TNF-α, IL-4, IL-6, IP-10, and Eotaxin than MtbCoM, which could diminish an oxidative response in LPSCoM-treated cells if these cells were exposed to high levels of cytokines that promote heightened activation and glycolytic metabolism.

The study reveals that MΦ-CM from Mtb- or LPS-stimulated hMDMs activates neutrophils, enhances their energetics, and influences their effector function, while also obstructing their ability to migrate away from the infection site (Figure 8). Recent pre-clinical work targeting neutrophils in animal models has revealed promising preliminary results. Targeting neutrophils with disulfiram, an FDA-approved drug for alcohol use disorder, reduces NET formation and perivascular fibrosis in hamster lungs, and downregulates innate immune and complement/coagulation pathways, suggesting that it could be beneficial for patients with COVID-19 in follow up studies [53]. Our data highlights the importance of the ongoing work in macrophage models of infection, as the development of anti-inflammatory medications is likely to ameliorate any downstream effects on neutrophils. Due to its effects on neutrophil activation, CXCL8 is likely to play a key role in our model; however, further studies are needed to delineate neutrophil function in this model [54]. Moreover, soluble mediators produced by activated macrophages may work synergistically to activate neutrophils. Considering the role neutrophils play in the progression of many diseases, understanding how an inflammatory microenvironment affects neutrophils is likely to aid the tailoring and design of new therapies for a range of infectious and inflammatory diseases.

### Study Limitations

In this study, we carried out an in-depth analysis of primary human neutrophil phenotype and function following exposure to conditioned medium from stimulated human macrophages. While we carefully designed our experiments to include rigorous controls, there are several limitations to this model due to the nature of working in vitro with primary human neutrophils. Human neutrophils by their nature are short-lived in vivo [55,56,57,58] and even shorter ex vivo making these experiments technically challenging. Therefore, the length of the assays is limited to ensure the viability of the neutrophils throughout. We used a one-hour incubation of neutrophils with MΦ-CM since a one-hour treatment has previously been established to be sufficient to induce changes in ROS production in human neutrophils [59]. We acknowledge that longer time points may be useful but are difficult to achieve in vitro with primary human neutrophils.

Whilst every effort was taken to remove any residual iH37Rv from the MtbCoM, using centrifugation and filtering, LPS was not removed from LPSCoM in the same way. While neutrophils are generally unresponsive to the bacterial species used as a source of LPS in the study (*E. coli* O55:B5), and only modestly express CD14 and TLR [40,60], we cannot rule out that residual LPS in the LPSCoM may have a direct effect on neutrophils in our model. However, we do not report profound differences between the effects of LPSCoM and MtbCoM on neutrophils. The LPSCoM contained higher amounts of cytokines, for example TNF-α, IFN-γ, and IL-6, and induced higher CD63 MFI, a marker of neutrophil activation, as well as higher production of IL-6 from the neutrophils compared with MtbCoM. In contrast, the MtbCoM had a much more profound effect on the OCR of the neutrophils following stimulation with fMLP. Overall, the effects of Mtb- or LPSCoM on neutrophils were mostly consistent, indicating that the effects of the microenvironment created by activated macrophages on neutrophil function are consistent regardless of the infection or stimulation. These similarities suggest that the macrophage–neutrophil axis described herein may apply to many settings of inflammation, not just infection with Mtb or Gram-negative bacteria.

We acknowledge that the work described herein is observational in nature and does not explore specific soluble mediators that are mechanistically responsible for inducing the effects on neutrophil function. However, this work is hypothesis-generating for future studies whereby blocking or neutralizing antibodies can further elucidate the key soluble factors propagating inflammation. This work is warranted to translate these findings toward therapeutic benefit. We also acknowledge that the MΦ-CM may contain other soluble factors that were not specifically quantified in this study but may nevertheless impact neutrophil function such as ROS or metabolites. Factors such as these can move through the transwell system used to assess neutrophil chemotaxis and influence neutrophil activation [61]. Additionally, as hMDM purity on the day of stimulation was approximately 95%, we cannot rule out the possibility of soluble mediators produced by small contaminating populations contributing to the cytokines present in the MΦ-CM.

## 4. Materials and Methods

### 4.1. Obtaining MΦ-CM for Functional Neutrophil Analysis

Peripheral blood buffy coats were obtained with consent from the Irish Blood Transfusion Services in Dublin, Ireland. PBMCs were isolated by density gradient centrifugation with Lymphoprep^TM^ (Stemcell Technologies, Vancouver, BC, Canada) and seeded at 2.5 × 10^6^ cells/mL in Roswell Park Memorial Institute (RPMI) 1640 medium (Bio-Sciences, Nottingham, UK), supplemented with 10% AB-human serum (Sigma-Aldrich, St. Louis, MO, USA), and plated onto non-treated cell culture plates (Corning Inc., Corning, NY, USA). LabTeks^TM^ (Nunc, Roskilde, Denmark) were also seeded to determine the multiplicity of infection (MOI; see below section). To obtain hMDMs, the cells were adherence-purified over 7–10 days at 37 °C and 5% CO_2_ to allow differentiation prior to experimentation. Cells were washed every 2–3 days to remove non-adherent cells, retaining hMDMs which stick to the plate surface. On day 7, hMDMs were routinely greater than 95% pure, as determined by flow-cytometry-based analysis of CD14 and CD68 co-expression.

The irradiated H37Rv Mtb strain (iH37Rv) was gifted by BEI Resources. Mtb was centrifuged at 3000× *g* for 10 min and resuspended in RPMI 1640 medium. The suspension was passed 10 times through a 25-gauge needle and centrifuged at 100× *g* for 3 min to remove any bacterial clumps. The volume of bacterial suspension required for a given MOI was determined by treating macrophages with a range of volumes of resuspended Mtb. hMDMs in Labteks^TM^ were incubated with iH37Rv-Mtb for 3 h, washed with pre-warmed PBS to remove extracellular bacteria, and fixed with 2% paraformaldehyde (PFA) (Sigma-Aldrich, St. Louis, MO, USA) for 10 min. hMDMs were subsequently stained with Modified Auramine O stain and Modified Auramine O decolorizer (Scientific Device Laboratory, Des Plaines, IL, USA) followed by Hoechst 33242 (Sigma-Aldrich, St. Louis, MO, USA) to counterstain the nuclei. The cells were analyzed under an inverted fluorescent microscope (Olympus IX51, Olympus, Tokyo, Japan) to determine the average number of phagocytosed bacilli per cell and percentage of cells infected. The required volume of bacilli was determined, phagocytic variation was adjusted between donors to ensure the same MOI (1–10 bacilli/cell, 70% positivity approximately), and the calculated volume of resuspended iH37Rv-Mtb was added to the appropriate experimental wells. hMDMs were also left unstimulated or stimulated with LPS (Sigma-Aldrich, St. Louis, MO, USA; 100 ng/mL in parallel). Then, 3 h later, supernatants containing the extracellular bacteria were washed off with PBS and replaced with complete RPMI (cRPMI). Macrophages were incubated for a further 21 h (24 h total). The MΦ-CM (UsCoM, MtbCoM, and LPSCoM) was filtered using a 0.20 μm filter to remove any residual iH37Rv-Mtb, pooled using samples from three independent preparations (n = 3), and stored at −80 °C.

### 4.2. Determination of Cytokine and Chemokine Levels in MΦ-CM in Response to Mtb and LPS Stimulation

To determine the secreted levels of cytokines and chemokines in UsCoM, MtbCoM, and LPSCoM, the data was repurposed from a previous published study from our laboratory (Cahill et al., 2021 [9]). Specifically, all TB drug treatment groups were omitted, and the data was reprocessed, replotted, and re-analyzed to only include unstimulated and Mtb-stimulated data (as shown). hMDMs were differentiated and adherence-purified from PBMCs, stimulated with iH37Rv (see below) or LPS (100 ng/mL). Supernatants, collected at 24 h post-stimulation, were assayed to determine the concentrations of cytokines and chemokines according to the manufacturers’ instructions (Meso Scale Discovery MultiArray Technology, Rockville, MD, USA). The cytokines assessed included IL-1β, IL-6, IL-8, IL-10, TNF-α, IFN-γ, IL-12, IL-13, IL-2, and IL-4. The chemokines assessed included MCP-1, MCP-4, IP-10, MDC, MIP-1β, Eotaxin, Eotaxin-3, TARC, and MIP-1α. This enabled the secreted levels of these products to be determined in UsCoM, MtbCoM, and LPSCoM, in order to examine how this MΦ-CM affects neutrophil biology.

### 4.3. Isolation of Neutrophils

Fresh blood was obtained with informed consent from the haemochromatosis clinic, in St. James’ Hospital, Dublin. The study was reviewed and approved by the St. James’s Hospital and the Tallaght University Hospital Research Ethics Committee. Neutrophils were separated from PBMCs using density gradients of Percoll^TM^ (GE healthcare, Uppsala, Sweden) and dextran (Sigma-Aldrich, St. Louis, MO, USA) (all reagents were used at room temperature or just out of the fridge). The required volume of blood was transferred to a 50 mL falcon tube with an equal volume of pre-warmed 2% dextran, gently inverted 18–20 times, and left at room temperature for 30 min to allow for erythrocyte sedimentation. The supernatant (cloudy yellow layer with no RBCs) was removed from the blood/dextran solution, placed into 15 mL tubes, and centrifuged at 200× *g* for 7 min with no brakes. The supernatant was slowly removed with a pipette to avoid disturbing the neutrophil-rich pellet and was disposed of. Then, 100% Percoll™ solution was made (9 parts Percoll™ + 1 part 1.5M NaCl), the pellet was resuspended in 55% Percoll™ solution (in PBS without Ca^2+^/Mg^2+^), 65% Percoll™ solution was carefully layered on top of this 55% Percoll™–neutrophil layer, and the layers were centrifuged at 1500× *g* for 30 min with no brakes. The supernatant was removed carefully and disposed of, and the cells were washed with the same PBS by centrifugation at 400× *g* for 5 min with the brakes on. Red blood cells were then removed using red blood cell lysis buffer made on-site (10×: 0.1 mM EDTA, 105.5 mM ammonium chloride, and 11.9 mM sodium bicarbonate, made up in dH_2_O; pH = 7.4) by resuspending the resulting pellet in 1× RBC lysis buffer for 5 min. The cells were then washed as before with PBS (Sigma-Aldrich, St. Louis, MO, USA) and centrifuged at 400× *g* for 5 min (brakes on). The cell pellet was then resuspended in 1 mL PBS, Seahorse media, or FACs buffer (PBS + 1% FBS) as required depending on the downstream application. To assess transwell migration, apoptosis, and neutrophil cytokine production, the neutrophils were isolated using the negative selection EasySep^TM^ Direct Human Neutrophil Isolation Kit (Stemcell Technologies, Vancouver, BC, Canada). Once isolated, the neutrophils were resuspended in RPMI supplemented with 1% FBS and counted. All neutrophils were isolated on the same day of experimentation (unless otherwise stated). Neutrophils were routinely greater than 95% pure, as determined by flow-cytometry-based analysis of CD15 expression.

### 4.4. Neutrophil Conditioning Using MΦ-CM

Freshly isolated neutrophils were seeded at a density of 2.5 × 10^5^ in each well of a 48 well plate. Cells were treated with 20% conditioned media, under 4 different conditions as follows: cRPMI (RPMI + 10% human serum), MΦ-CM from unstimulated hMDMs (UsCoM), MΦ-CM from Mtb-stimulated hMDMs (MtbCoM), or MΦ-CM from LPS-stimulated hMDMs (LPSCoM). Following one hour of incubation, the conditioned media was washed off and the cells were resuspended in fresh cRPMI supplemented with 1% FBS. Neutrophils were artificially activated using 1.25 ng/mL N-Formylmethionine-leucyl-phenylalanine (fMLP; Sigma-Aldrich, St. Louis, MO, USA), where applicable.

### 4.5. Stimulation of Neutrophils with Mtb

iH37Rv was processed as previously stated. Neutrophils were seeded in 8-well Labteks^TM^ (pre-coated with undiluted human serum for one hour), incubated with iH37Rv-Mtb for 1 h, washed with pre-warmed PBSto remove extracellular bacteria, and fixed with 2% paraformaldehyde for 10 min. Neutrophils were subsequently stained with Auramine and Hoechst 33242, analyzed, and an MOI was determined as per above. The calculated volume of iH37Rv-Mtb was added to the appropriate experimental ports of a Seahorse XFe24 Analyzer (Seahorse Biosciences, North Billerica, MA, USA) for neutrophil metabolism analysis.

### 4.6. Characterizing the Effect of MΦ-CM on Neutrophil Metabolism Utilizing the Seahorse XFe24 Analyzer

Neutrophils were seeded at 2.5 × 10^5^ cells per well in a 24-well cell culture XF microplate using Poly-L-lysine (Seahorse Biosciences, North Billerica, MA, USA). Neutrophils were incubated for one hour with cRPMI (10% HS), UsCoM, MtbCoM, or LPSCoM, and washed with assay medium (Seahorse XF medium, supplemented with 14.6 mg L-glutamine, 500 µL glucose, and 500 µL pyruvate) (Agilent, Santa Clara, CA, USA) before incubation with assay medium for 30 min at 37 °C in a non-CO_2_ incubator. OCR and ECAR, reflecting OXPHOS and glycolysis, respectively, were measured. Four baseline OCR and ECAR measurements were obtained over 20 min. Nineteen subsequent OCR and ECAR measurements were also obtained over 110 min following treatment with fMLP or Mtb.

### 4.7. Characterizing the Effect of MΦ-CM on Neutrophil Effector Functions Using Flow Cytometry

To assess the effect of the MΦ-CM on neutrophil effector functions, flow cytometry was used. To determine ROS production, DHR123 (Invitrogen, Paisley, UK) was used. Following isolation, 5 × 10^5^ neutrophils were seeded in FACS tubes (Corning Inc., Corning, NY, USA) in RPMI + 1% FBS and then loaded with 5 μg/mL Cytochalasin B (Sigma-Aldrich, St. Louis, MO, USA) and 2.5 μg/mL DHR123. The tubes were incubated in the dark for 10 min at 37 °C and were then stained with Zombie Aqua Fixable Viability Dye (BioLegend, San Diego, CA, USA). The neutrophils were then treated with 20% cRPMI, UsCoM, MtbCoM, or LPSCoM for one hour at 37 °C. Following this, 1% PFA (Sigma-Aldrich, St. Louis, MO, USA) was added for 15 min at room temperature to stop the stimulation. The tubes were then centrifuged for 5 min at 300× *g* to pellet the cells and washed twice. Cells were resuspended in PBS + 1% FBS and stained with CD15 (PE-Cy7; BioLegend, San Diego, CA, USA) in the dark for 10 min at room temperature. Cells were then washed with PBS + 1% FBS. To measure NETosis, 5 nM Sytox Red (Invitrogen, UK) and 0.3 nM DAPI (Sigma-Aldrich, St. Louis, MO, USA) diluted in PBS + 1% FBS was added, and the tubes were incubated in the dark for 15 min at room temperature. Samples were then immediately acquired on a BD FACS Canto II. To measure glucose uptake, 2-NBDG (Invitrogen, UK) was used. Following isolation, 5 × 10^5^ neutrophils were seeded in FACS tubes in RPMI + 1% FBS and then loaded with 100 μM 2-NDBG and then treated with 20% cRPMI, UsCoM, MtbCoM, or LPSCoM for one hour at 37 °C. Then, 1% PFA was added for 15 min at room temperature to stop the stimulation. The tubes were then centrifuged for 5 min at 300× *g* to pellet the cells and washed twice. Cells were resuspended in PBS + 1% FBS and stained with CD15 (PE-Cy7) in the dark for 10 min at room temperature. Neutrophils were then washed with PBS + 1% FBS and immediately acquired on a BD FACS Canto II (BD, Franklin Lakes, NJ, USA)

To investigate the effect of the MΦ-CM on neutrophil activation, migration, and apoptosis, neutrophils were seeded and treated as described above. Neutrophils were then centrifuged at 300× *g* for 5 min and resuspended in PBS + 1% FBS (or Annexin V Binding Buffer for apoptosis assays) and stained with Zombie Aqua Fixable Viability Dye, CD15 (PE-Cy7), CD62L (BV421; BioLegend, San Diego, CA, USA), CD63 (PE; BioLegend, San Diego, CA, USA), CXCR2 (APC; BioLegend, San Diego, CA, USA), or Annexin V (PE; BioLegend, San Diego, CA, USA) in the dark for 10 min at room temperature. Cells were then resuspended in PBS + 1% FBS (or Annexin V Binding Buffer for apoptosis assays), centrifuged at 300× *g* for 5 min, and immediately run on a BD FACS Canto II. Unstained cells and FMO controls were used to normalize for background staining and to set the gates. Data were analyzed using the FlowJo^TM^ software v10.

### 4.8. Neutrophil Transwell Assay and Quantification Using Counting Beads

To assess the effect of the conditioned media on neutrophil chemotaxis, transwell assays were performed. Firstly, 5 µm pore 24-well tissue culture inserts were coated (Sarstedt, Nümbrecht, Germany) with undiluted human serum for 1 h (the human serum enables neutrophil adhesion). The serum was washed off once using PBS and inserts were left in the hood to dry for 1 h. RPMI 1640 (1% FBS, 1.25 ng/mL fMLP) was added to the bottom chamber. Once dry, 5 × 10^5^ conditioned-media-stimulated neutrophils were added to the top chamber. Cells were placed into an incubator and allowed to migrate for 15 min, after which, the inserts were removed. Cells in the bottom chamber were quantified using flow cytometry. Following the transwell assay, neutrophils in the bottom chamber were transferred to FACS tubes and stained using CD15 (PE-Cy7; BioLegend, San Diego, CA, USA)) and a Live/Dead stain (Aqua, Invitrogen, UK) according to the staining procedure described above. Then, 50 µL of precision count beads (BioLegend, San Diego, CA, USA) at a concentration of 1000 beads/µL were added to each tube and the samples were acquired using BD FACS Canto II. Relative cell counts [cells/µL] were calculated by diving the cell event count by the precision count beads event count and multiplying this value by the precision count beads concentration (beads/µL). Absolute cell counts could be counted by multiplying the relative cell count by the volume of the sample (in µL).

### 4.9. Neutrophil Multiplex ELISAs

To assess the effect of the MΦ-CM on neutrophil inflammatory cytokine production, the Meso Scale Discovery (MSD) multiplex ELISA assay was employed (Meso Scale Discovery MultiArray Technology, Rockville, MD, USA). Freshly isolated neutrophils were treated for 1 h with cRPMI, UsCoM, MtbCoM, and LPSCoM, washed off following incubation, and resuspended in fresh cRPMI. Neutrophils were then either left unstimulated or stimulated with 1.25 ng/mL fMLP, to examine if subsequent stimulation changed cytokine levels further. Supernatants were collected 4 h later and screened for the levels of IL-1β, IL-6, IL-8, IL-10, TNF-α, IFN-γ, IL-12, IL-13, IL-2, and IL-4 according to the manufacturers’ instructions.

### 4.10. Statistical Analysis

Data were analyzed using GraphPad Prism software version 9 (GraphPad Prism, San Diego, CA, USA). Friedman ANOVA tests with Dunn’s multiple comparisons tests were utilized to statistically analyze differences in datasets with three or more groups. Two-way RM ANOVA with Šídák’s multiple comparisons tests were used to statistically analyse Seahorse time course data (Figure 5D,E). Two-way RM ANOVA with mixed effect multiple comparisons tests were used to statistically analyze grouped MSD data (Figure 6B). Differences of *p* < 0.05 (*), *p* < 0.01 (**), *p* < 0.001 (***), and *p* < 0.0001 (****) were considered statistically significant.

## 5. Conclusions

Our findings underscore the dual nature of neutrophils, acting as frontline defenders yet potentially contributing to tissue damage and mortality in certain scenarios. The research highlights the impact of MtbCoM and LPSCoM on neutrophil activation, metabolism, and function. Interestingly, our results suggest that, despite the pre-treatment with MΦ-CM, neutrophils responded differently when directly infected with Mtb, suggesting a pathogen-specific adaptation in their metabolic responses. These findings underscore the complexity of neutrophil responses to different stimuli as well as contribute valuable insights into the macrophage–neutrophil axis, unraveling potential therapeutic targets and paving the way for further investigations into the complexities of neutrophil responses in various disease contexts. As macrophage–neutrophil interactions prove crucial in diverse pathological conditions, understanding these interactions holds promise for the development of tailored therapies in the fight against infectious and inflammatory diseases.

## Figures and Tables

**Figure 1 ijms-25-02898-f001:**
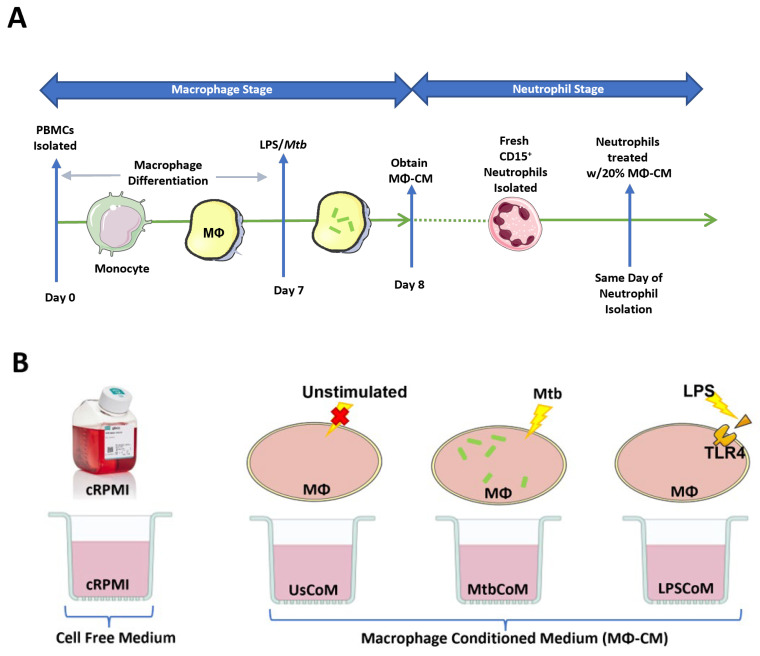
Methods Summary. (**A**) Macrophage Stage: hMDMs, differentiated from PBMCs isolated from healthy blood donors over 7 days, were stimulated with Mtb for three hours and washed to remove unphagocytosed Mtb (hMDMs were also left unstimulated and treated with 100 ng/mL LPS in parallel). Then, 24 h post-stimulation, the hMDM-conditioned medium (MΦ-CM) was processed and stored at −80 °C for later use. MΦ-CM from a minimum of three independent donors was pooled and was used for all subsequent experimentation techniques. Neutrophil Stage: On a subsequent day, fresh CD15+ cells were isolated, purified, and treated with 20% MΦ-CM (80% fresh cRPMI) for one hour. Flow cytometry, real-time extracellular Seahorse flux analysis, and MSD ELISAs were subsequently carried out to determine the effect of the MΦ-CM on neutrophil function. (**B**) All treatments with MΦ-CM consisted of freshly isolated neutrophils being exposed to 20% cRPMI (RPMI supplemented with 10% human serum), 20% MΦ-CM from unstimulated hMDMs (UsCoM), 20% MΦ-CM from Mtb-stimulated hMDMs (MtbCoM), and 20% MΦ-CM from LPS-stimulated hMDMs (LPSCoM).

**Figure 2 ijms-25-02898-f002:**
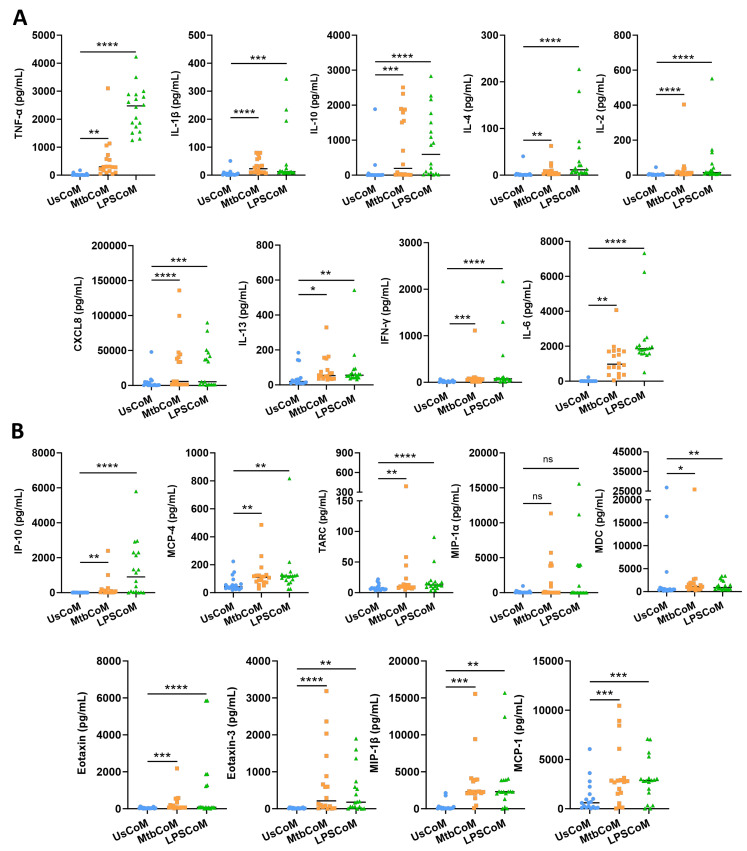
Characterizing secreted cytokine and chemokine levels in MΦ-CM (UsCoM, MtbCoM, and LPSCoM) from unstimulated, Mtb-stimulated, and LPS-stimulated hMDMs. hMDMs, differentiated from PBMCs isolated from healthy blood donors, were stimulated with iH37Rv Mtb for three hours and were washed to remove unphagocytosed Mtb. hMDMs were also left unstimulated and treated with LPS (100 ng/mL) in parallel. (**A**) 24 h post-stimulation, the secreted levels of the cytokines TNF-α, IL-1β, IL-10, IL-4, IL-2, CXCL8, IL-13, IFN-γ, and IL-6 were quantified using MSD Multi-Array technology, (**B**) along with the chemokines IP-10, MCP-4, TARC, MIP-1α, MDC, Eotaxin, Eotaxin-3, MIP-1β, and MCP-1. Bars denote mean ± SEM (n = 18 independent donors). * *p* < 0.05, ** *p* < 0.01, *** *p* < 0.001, and **** *p* < 0.0001, ns = no significance (Friedman ANOVA test with Dunn’s multiple comparisons tests).

**Figure 3 ijms-25-02898-f003:**
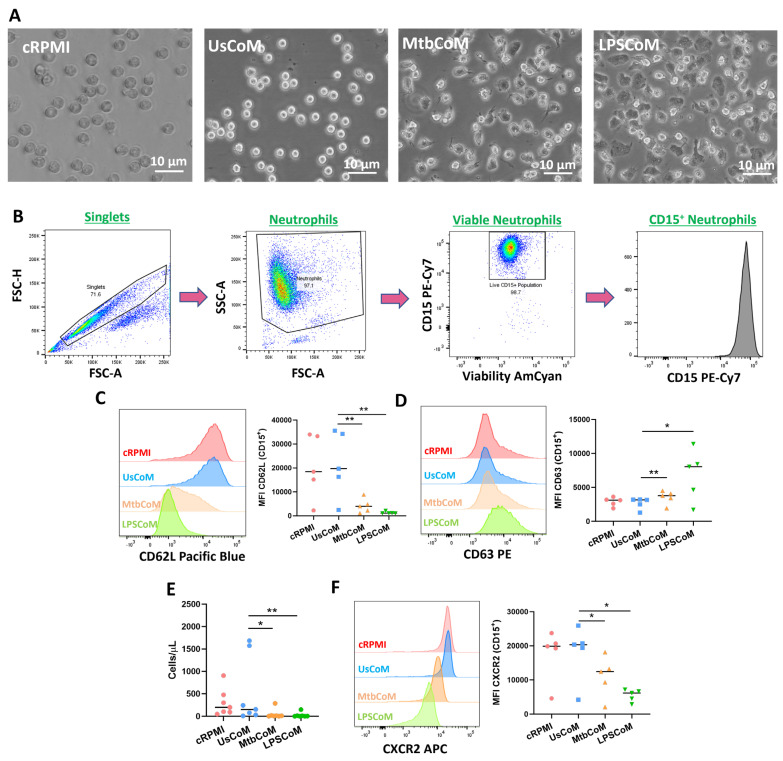
Examining if MΦ-CM from Mtb-stimulated or LPS-stimulated hMDMs affects human neutrophil activation and migration. (**A**) Neutrophils exposed to 20% MΦ-CM were visualised under light microscopy (40×) to examine if the MΦ-CM affects neutrophil morphology. (**B**) Flow cytometric analyses were carried out to examine if neutrophils exposed to MΦ-CM could be activated, first by gating on single cells (FSC-A vs. FSC-H), granular cells (FSC-A vs. SSC-A), and finally viable CD15^+^ neutrophils (AmCyan vs. PE-Cy7). (**C**) Neutrophil activation was assessed by determining the expression of the granulation marker CD62L and (**D**) the degranulation marker CD63 in viable CD15^+^ neutrophils in response to cRPMI, UsCoM, MtbCoM, and LPSCoM (n = 5). (**E**) As activation status is a good surrogate indicator of migration in neutrophils, the ability of neutrophils to migrate was also assessed. Cells were treated with 20% MΦ-CM for one hour as before, placed into the top chamber of 5 µm transwell inserts, and allowed to migrate toward 1.25 ng/mL fMLP in the bottom chamber for 15 min. Samples were then stained for CD15 expression and viability, an equal number of Precision^TM^ Counting Beads were added, and the numbers of migrated neutrophils (cells/µL) were quantified by flow cytometry (n = 6). (**F**) Expression of the cell surface migration marker CXCR2 was examined in neutrophils exposed to the MΦ-CM treatments (n = 5). * *p* < 0.05 and ** *p* < 0.01 (Friedman ANOVA test with Dunn’s multiple comparisons tests). MFI: Median fluorescence intensity.

**Figure 4 ijms-25-02898-f004:**
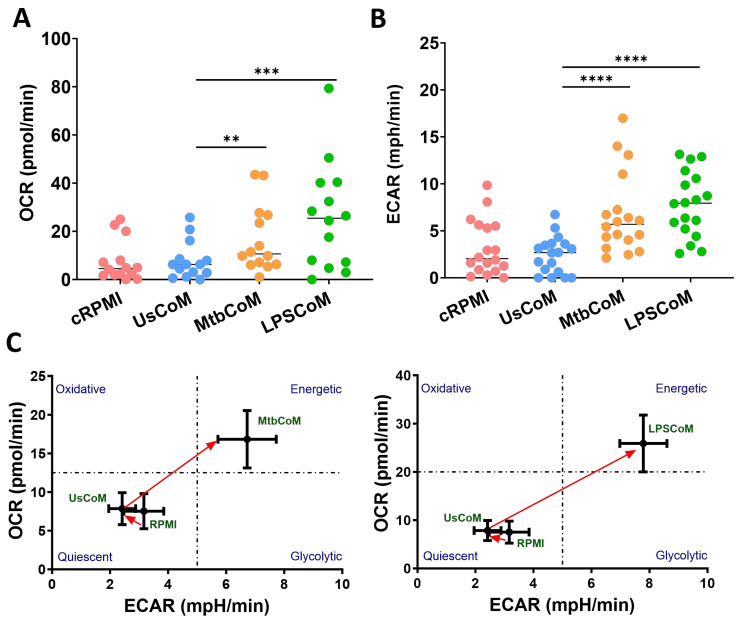
Investigating if MΦ-CM from Mtb-stimulated or LPS-stimulated hMDMs affects baseline glycolysis and oxidative phosphorylation in unstimulated human neutrophils. The effect of MΦ-CM on the real-time baseline (**A**) OCR (n = 14) and (**B**) ECAR (n = 18), representing oxidative phosphorylation and glycolysis, respectively, was determined utilising Seahorse extracellular flux assays in neutrophils treated for one hour with 20% cRPMI, UsCoM, MtbCoM, and LPSCoM. (**C**) The immunometabolic shift from UsCoM treatment to MtbCoM and LPSCoM treatment can be illustrated by the metabolic phenograms. ** *p* < 0.01, *** *p* < 0.001, and **** *p* < 0.0001 (Friedman ANOVA test with Dunn’s multiple comparisons tests).

**Figure 5 ijms-25-02898-f005:**
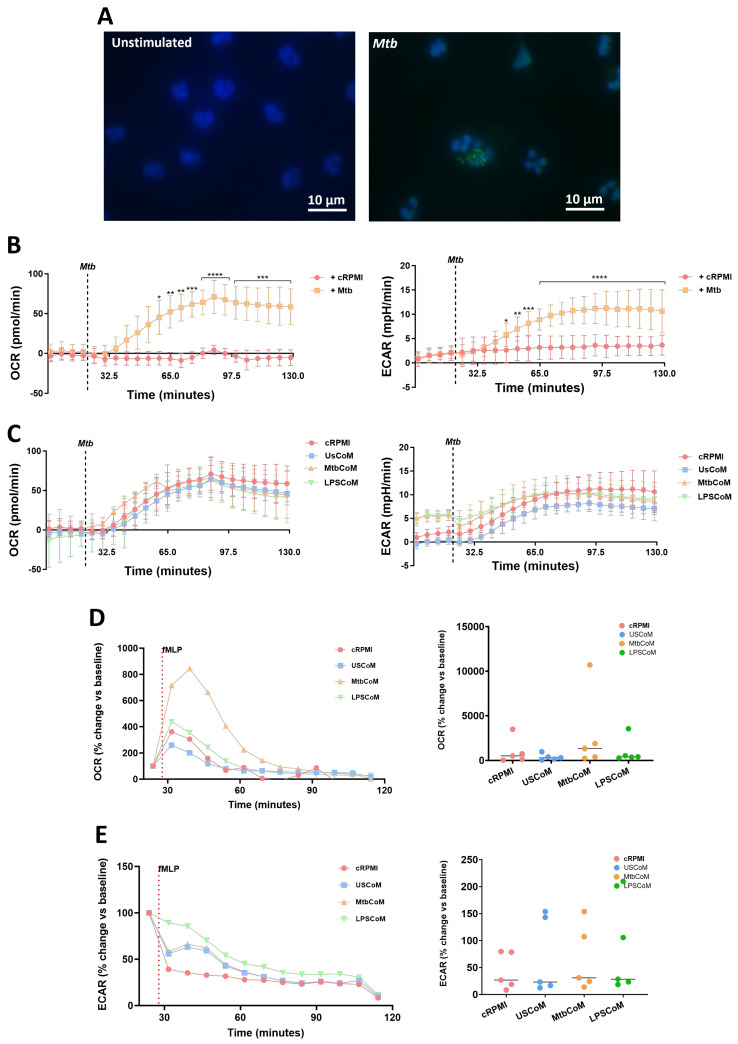
Assessing if MΦ-CM from Mtb-stimulated or LPS-stimulated hMDMs affects bioenergetics in fMPL-stimulated and Mtb-stimulated human neutrophils. (**A**) Neutrophils were seeded into Labtek II chamber slides, unstimulated or stimulated with iH37Rv-Mtb, washed, and phagocytosed Mtb was fluorescently visualised through auramine (FITC) and hoechst (DAPI) staining. (**B**) Baseline metabolic OCR (left) and ECAR (right) values were examined in neutrophils in response to iH37Rv-Mtb (MOI: 1–10), (**C**) prior to determining if 20% cRPMI, UsCoM, MtbCoM, and LPSCoM pre-treatment exhibited an effect on OCR and ECAR profiles in neutrophils stimulated with iH37Rv (n = 3). After neutrophils were treated for one hour with 20% cRPMI, UsCoM, MtbCoM, and LPSCoM, cells were also stimulated with fMLP (1.25 ng/mL) and real-time baseline (**D**) OCR and (**E**) ECAR values were determined utilising Seahorse extracellular flux assays (n = 5). (**D**,**E**) OCR and ECAR rates were plotted and graphed (right-hand side graphs) at the time point reflecting the highest metabolic response to fMLP stimulation (40 min). * *p* < 0.05, ** *p* < 0.01, *** *p* < 0.001, and **** *p* < 0.0001 ((**A**,**B**): Friedman ANOVA test with Dunn’s multiple comparisons tests; (**D**,**E**): Two-way RM ANOVA with Šídák’s multiple comparisons test).

**Figure 6 ijms-25-02898-f006:**
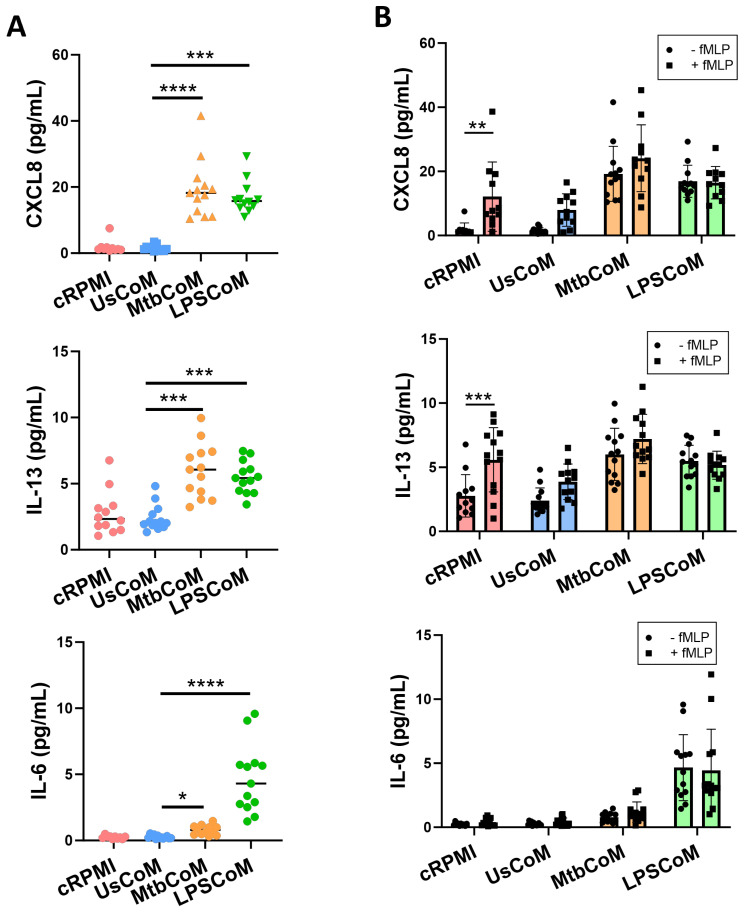
Elucidating if MΦ-CM from Mtb-stimulated or LPS-stimulated hMDMs alters cytokine release in primary human neutrophils. Human neutrophils were treated with 20% cRPMI, UsCoM, MtbCoM, and LPSCoM for 1 h. Following incubation, neutrophils were left unstimulated (**A**) and stimulated with 1.25 ng/mL fMLP (**B**). 4 h later, the secreted levels of CXCL8, IL-13, and IL-6 were assessed using MSD-ELISA (n = 13). * *p* < 0.05, ** *p* < 0.01, *** *p* < 0.001, and **** *p* < 0.0001 ((**A**): Friedman ANOVA test with Dunn’s multiple comparisons tests; (**B**): Two-way ANOVA with mixed effects model multiple comparisons test).

**Figure 7 ijms-25-02898-f007:**
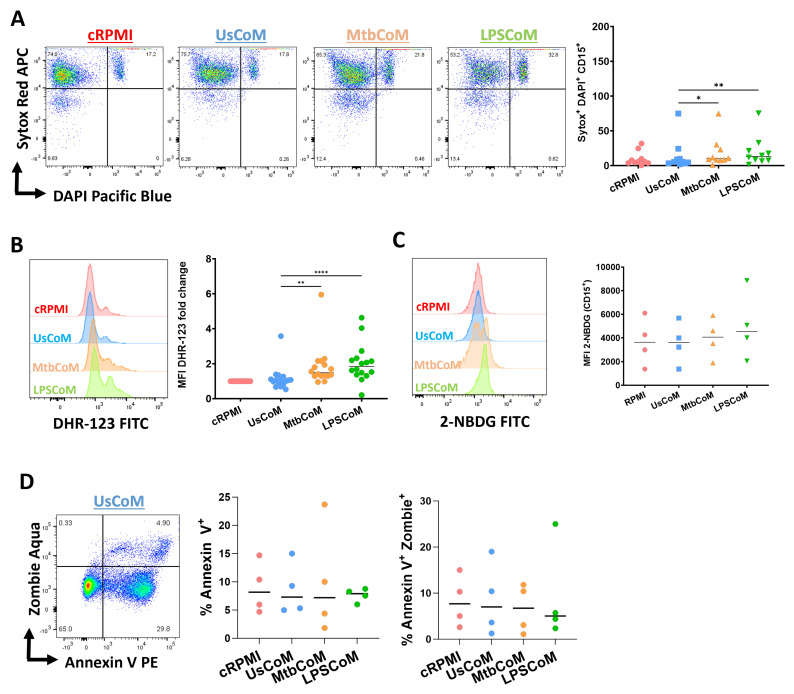
Determining if MΦ-CM from Mtb-stimulated or LPS-stimulated hMDMs alters NETosis, reactive oxygen species (ROS) production, glucose uptake, and apoptosis in primary human neutrophils. Flow cytometric analyses were carried out to examine if neutrophils exposed to MΦ-CM exhibited altered NETosis, ROS production, and glucose uptake. This was achieved by first gating on single cells (FSC-A vs. FSC-H), granular cells (FSC-A vs. SSC-A), and finally, viable CD15+ neutrophils (AmCyan vs. PE-Cy7). (**A**) Flow cytometric dot plots illustrating the effect of 20% MΦ-CM on NETosis in a representative donor (NETosis is characterized as being double positive in the Sytox red (APC) and DAPI (Pacific Blue) channels). Histogram shifts in (**B**) DHR-123 (FITC) and (**C**) 2-NBDG (FITC) expression are characterized by changes in ROS production and glucose uptake in MΦ-CM-treated CD15+ neutrophils, respectively. Using these flow cytometric tools, NETosis (n = 10), ROS production (n = 16), and glucose uptake (n = 4) were assessed in CD15+ neutrophils one hour post-treatment with 20% MΦ-CM. (**D**) Early apoptosis (characterized as CD15+/Annexin V+) and late apoptosis (characterized as CD15+/Annexin V+/Zombie+) were also determined to examine the effect of the MΦ-CM on cell viability. A representative flow cytometric dot plot is shown. * *p* < 0.05, ** *p* < 0.01, **** *p* < 0.0001 (Friedman ANOVA test with Dunn’s multiple comparisons tests).

**Figure 8 ijms-25-02898-f008:**
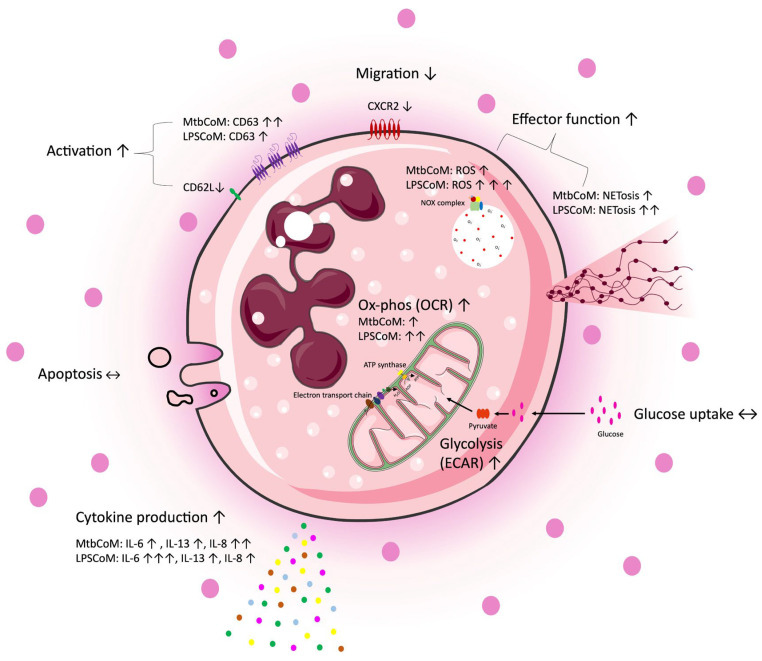
A graphical summary of the study. Freshly isolated CD15+ neutrophils were exposed to four different conditions, complete RPMI media (cRPMI), media from unconditioned MDMs (UsCoM), media from MDMs infected with Mtb (MtbCoM), and media from MDMs stimulated with LPS (LPSCoM). Glucose uptake and apoptosis were unaffected by MtbCoM and LPSCoM. Effects on several aspects of neutrophil function were examined via several methods, including flow cytometry, Agilent Seahorse assays, MSD ELISA, and transwell migration assays. Our data suggests an augmentation of many neutrophil functions as a result of MtbCoM and LPSCoM, such as NETosis, ROS generation, and inflammatory cytokine production. Neutrophils exposed to the inflammatory conditioned media also exhibited a significant energetic shift, with glycolysis and oxidative phosphorylation significantly augmented. MtbCoM and LPSCoM also decrease neutrophil chemotaxis, as shown both via the quantification of the chemotactic receptor CXCR2 and via transwell migration assays. Overall, our data demonstrated the importance of the macrophage–neutrophil axis in infection.

## Data Availability

Data are contained within the article and Appendix A.

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
