# Peer review of "Human Macrophages Activate Bystander Neutrophils’ Metabolism and Effector Functions When Challenged with Mycobacterium tuberculosis"

_ijms, 2024, doi:10.3390/ijms25052898_

Round 1

Reviewer 1 Report (Previous Reviewer 2)

Comments and Suggestions for Authors

The revised version contains some added details in the experimental as requested. i don(t see more in the Discussion, I may have missed, but I don't see much more insight in the Discussion.  

Author Response

Reviewer 1 comments: The revised version contains some added details in the experimental as requested.

Response to reviewer 1 comments: Thank you for your continued engagement with our manuscript and for providing your insightful feedback during the review process. We appreciate your diligence in assessing the changes made to address your previous comments.

Upon re-evaluation, we acknowledge that while we have addressed some aspects of your previous queries there may still be room for further elaboration and depth in the Discussion section. In response to your comments, many of the sections discussed prior have been revised, more specifically removing any repetition between the ‘results’ and ‘discussion’ sections, and making the ‘discussion’ much more concise. We have expanded on certain ideas, such as the stage of infection in our model (lines 461-482), and the potential role of specific cytokines/chemokines we found upregulated in or CoMs (lines 319-350)/as a result of neutrophil priming (lines 433-445). We also touched on the potential pathways to examine in the future of this work based off our current observations, such as of looking further into CXCL1 which was not examined in our study but may be present in our CoMs due to the observation of effects on CXCR2 (lines 378-387). Additionally, we streamlined the discussion on our findings on the metabolic state of neutrophils post CoM treatment, focusing on the key findings (lines 388-430). The revised version also now emphasizes that MtbCoM or LPSCoM boosts the metabolic profile of unstimulated neutrophils, but subsequent infection with Mtb overrides any effect of pre-treatment with MΦ-CM, a novel finding. We've clarified the observed increase in oxygen consumption after fMLP stimulation in MΦ-CM pre-treated cells, attributing it to the heightened oxidative burst (lines 405-430). We hope these changes have given additional insight into our key finding in the discussion section.

Moreover, we restructured our discussion, ensuring paragraphs in the discussion section to correspond with the order of presentation of results in the manuscript While not reducing the word count of the discussion, these efforts contribute to a more efficient and succinct presentation of our research findings and we hope you find this version of the discussion much more insightful.

Reviewer 2 Report (Previous Reviewer 5)

Comments and Suggestions for Authors This revised version is, in my opinion, fully acceptable.

Author Response

Reviewer 2 comments: This revised version is, in my opinion, fully acceptable. 

Response to reviewer 2 comments: Thank you for taking the time to review the revised version of our manuscript. We are delighted to hear that you find the revised version fully acceptable. Your positive evaluation is greatly appreciated and reinforces our efforts to address the previous concerns raised during the review process.

Reviewer 3 Report (New Reviewer)

Comments and Suggestions for Authors

In their manuscript titled ` Human macrophages activate bystander neutrophil metabolism and effector function when challenged with Mycobacterium Tu-3 berculosis`, Murphy et al., have investigated the impact of macrophage-neutrophil axis in aggravating neutrophil response in acute infectious conditions as can be witnessed in active TB. Using conditioned medium from activated macrophages (MΦ-CM), the authors have explored several neutrophil responses (activation, degranulation, ROS generation, apoptosis, NETosis) and arrived at the conclusion that MΦ-CM from stimulated hMDMs can activate neutrophils, change their metabolic status, boost their inflammatory functions, and by reducing migratory potential sequester them at sites of infection. The manuscript is well drafted describing this straight forward study, offers essential details on the methological aspects and also points out the limitations of the study. The graphical abstract is well done and summarizes the results at a glance. Although the investigation has its merits and will be of interest to the readers, there are a few critical aspects that must be considered-

1.      A characterization of the hMDMs is currently lacking except for the inflammatory mediators profile (both inflammatory mediators and IL-10 significantly upregulated) and indication (data not shown) that they are CD14+CD68+ hMDMs. The authors need to characterize the hMDMs if they are predominantly of the M1 or M2 subtype, which will add vital information for the investigation.

2.      The proper control for LPSCoM media should have been RPMI complete media supplemented with LPS at the same concentration, since it is not clear from the methods section if LPS was removed/ added to collect LPSCoM, 21hrs later. Residual LPS in LPSCoM, could directly cause neutrophil activation and account for the observed changes.

3.      Please show individual data points for all the graphs throughout the manuscript. This is particulary evident judging by the lack of statistical significance in Figure 2B for MIP-1β.

4.      Since the authors noticed a downregulation in the surface expression of CXCR2, it will be worthwhile exploring the levels of CXCL1, CXCL8 in the MtbCoM and LPSCoM as compared to UsCoM.

5.      Did the neutrophils show migration towards UsCoM, MtbCoM and LPSCoM? Data on this may reveal the reason behind lack of subsequent neutrophil migration in response to fmlp.

6.      The markers of NETosis used in this study are not convincing enough, besides neutrophils need to undergo an incubation period of 4 hrs to show signs of NETosis. This experiment must be repeated with extended period of neutrophil incubation with UsCoM, MtbCoM and LPSCoM. The authors can use CitH3 levels, extracellular neutrophil elastase/ MPO activity of neutrophil supernatant as NETosis markers or visualize NETs with Sytox green.

7.      The investigators have used the Seahorse XFe24 platform to estimate OCR and ECAR, which reflects mitochondrial metabolism but ECAR does not always correspond to glycolysis. Please specify the test kit used in this study and the concentrations of mitochondrial toxins. To specifically pinpoint the preferential use of metabolic fuels the Seahorse XF Mito Fuel Flex Test kit could be used. Please provide data on the following parameters- ATP production, basal respiration, coupling efficiency, maximal respiration, nonmitochondrial respiration, spare respiratory capacity; and provide the entire profiles of OCR and ECAR in response to modulators of respiration (oligomycin, FCCP, and rotenone/ antimycin A).

Author Response

Thank you for taking the time to review our manuscript. Responses to your comments are attached as PDF.

Reviewer 4 Report (New Reviewer)

Comments and Suggestions for Authors

Minor points

- I miss a more detailed analysis of the interaction IL6

- According to lines 87 and 88, the blood sample was taken from a healthy person; Why was not done with three repetitions, that is, with 3 blood samples from different healthy people?

- It would not be better to use a positive blood sample from a patient with tuberculosis as a positive control?

Comments on the Quality of English Language

Major points

1.      Several spelling and grammatical errors in manuscript such as lines 93-94 page 3 four "And" in one sentence,  line 96 “consists”, line 601 “2.5 × 105”  do you mean 105, line 22 "effect" or affect

2. Improve writing structure”

Author Response

Thank you for taking the time to review our manuscript. The responses to your comments are attached as PDF.

Round 2

Reviewer 3 Report (New Reviewer)

Comments and Suggestions for Authors

the authors have responded to the issues raised during previous review

This manuscript is a resubmission of an earlier submission. The following is a list of the peer review reports and author responses from that submission.

Round 1

Reviewer 1 Report

Comments and Suggestions for Authors

I consider this paper interesting, scientifically valuable and informative, it contributes to better understanding of intercellular communication and represents a valuable basis for future research. Objections are primarily considering manuscript writing. In the last paragraph of the Introduction section, it is necessary to add only the aim of the research, not the results and methodology. The methodology should not be stated in the first paragraph of the Results section, because there is a section Materials and methods for that, where the same is stated. The discussion should be shortened, avoid repetition of already stated results. Focus on the explanation of the obtained results and their comparison with other studies.

Author Response

Thank you very much for your thorough review of our manuscript. We are very pleased that you found the research valuable and informative. We have now reviewed your comments and have responded accordingly below.

Reviewer 1 Comment 1: ‘In the last paragraph of the Introduction section, it is necessary to add only the aim of the research, not the results and methodology’.

Author’s Response to Reviewer 1 Comment 1: Thank you for this suggestion, we have now implemented suggested changes (lines 63-72).

Reviewer 1 Comment 2: The methodology should not be stated in the first paragraph of the Results section, because there is a section Materials and methods for that, where the same is stated.

Author’s Response to Reviewer 1 Comment 2: We have now tweaked this accordingly, removing mention of Methods from the first paragraph of Results section (lines 74-84).

Reviewer 1 Comment 3: The discussion should be shortened, avoid repetition of already stated results. Focus on the explanation of the obtained results and their comparison with other studies’.

Author’s Response to Reviewer 1 Comment 3:  Thank you for your feedback regarding this.  The  ‘Study Limitations’  subsection previously part of Discussion has now been made a separate section, shortening discussion by 472 words. Further, through editing, Discussion was shortened by a further 142 words.

Reviewer 2 Report

Comments and Suggestions for Authors

Author Response

Thank you very much for your thorough review of our manuscript. We are very pleased that you found the research valuable and informative. We have now reviewed your comments and have responded accordingly below.

Reviewer 2 Comment 1: ‘In terms of effector functions, the kinetics of neutrophils and macrophages are quite different: when neutrophils are activated, there is a short but intense burst of ROS while LPS- activated macrophages, for example, produce a sustained burst of ROS of much lower intensity. It is not clear whether you suggest here activation of lung-resident macrophages (alveolar) or macrophages associated with blood vessel, an interaction recently proposed to be important for neutrophil recruitment. Could you please be more specific? How would you describe the stage of infection your work models?

Author’s Response to Reviewer 2 Comment 1: Thanks for allowing us to clarify this. Many believe neutrophils are the first immune cells recruited to the lung during infection, by resident alveoloar macrophages in the lung, allowing them to rapidly respond to the invading pathogen (doi: 10.1038/nri3024). We specifically set out to establish a model to study the early effects of human macrophage and neutrophil interactions in an infectious disease setting. A publication (doi: 10.1016/j.cmet.2021.03.018) comparing neutrophil function and metabolism in acute and chronic inflammation also supported our hypothesis that the early priming of metabolic function by the inflammatory environment, underpins rapid neutrophil responses. They show that neutrophils undergo gluconeogenesis and glycogenesis in response to acute physiological stress and activation, and metabolic shifts that are required for fast energy production, function and survival. Our study further highlights and emphasizes the potency with which MtbCoM or LPSCoM can activate neutrophils and shows that an initial stimulatory or activation event could potentially lead to a rapid downstream cascade of activated neutrophils. We have detailed this study in the discussion section (lines 503-549).

Reviewer 2 Comment 2: In terms of secretory profile of the neutrophils, this works well describes the release of IL13, a cytokine known to shift the M1/M2 profile of macrophages toward M2 phenotype, while LPS is know to induce M1 macrophage phenotype. What about IL4? Could the authors please comment whether the cytokines release of the neutrophils affect the macrophages? The authors have used a transwell for a migration assay. Why not seeding macrophages in the bottom compartment and see if there is a bi-directionnal cross-talk? Did you determine the phagocytosis function ? Is it parallel with ROS release?

Author’s Response to Reviewer 2 Comment 2: Author’s Response to Reviewer 2 Comment 2: There is now ample evidence that IL4 can suppress the production of many monocyte and macrophage pro-inflammatory mediators, thereby promoting an M2 phenotype (DOI: 10.3389/fimmu.2014.00420). 

With regards to the cytokine release of the neutrophils and whether it affects macrophages, we believe this is likely too, although we can only speculate about this at this time considering this was outside the scope of the current study. The aim of the current study was to assess the specific effect of macrophage conditioned medium on neutrophil function. We simply did not have the resources or the budget to examine how the neutrophil affected neighbouring macrophages. The reviewer does raise a good subject, however. This was very evident when we conducted and published a literature review on the topic in late 2022 (DOI: 10.3389/fimmu.2022.984293). In this review, we discussed known interactions between neutrophils and other immune cells in an infectious disease setting and speculated what it could mean during active TB disease. It was very evident during this process that further research examining the link(s) between neutrophils and other immune cells (including macrophages) was warranted in the field. We subsequently set out to fill this research gap first by specifically examining how human macrophages (immune cells that our laboratory specialises in the most) affect human neutrophil function. Our aim in the future is to examine how various immune cells affect neutrophil function (and vice versa) in the human lung in the hope that we can learn more about the complex interactions and dynamics of these cells in vivo during lung infections. 

Reviewer 2 Comment 3: ‘Overall, it is sometimes difficult to follow the experimental details, which would be useful for the reader. Details should be given on the way you differentiate the monocytes. Usually, primary cells are quite fragile and seven days is a long time. Many protocols involve only three days in the presence of CSF1 or GM-CSF for example. Please comment’.

Author’s Response to Reviewer 2 Comment 3: Apologies that the process of describing the differentiation of our human macrophage model isn’t clear in our manuscript. For ~25 years now, our laboratory has worked on and published many times on this primary macrophage model (DOI: 10.1128/iai.65.1.298-304.1997, 10.1128/IAI.00614-07, 10.1165/rcmb.2010-0319OC, 10.1164/rccm.201407-1385OC, 10.4049/jimmunol.1501612, 10.1165/rcmb.2017-0382OC, 10.1165/rcmb.2018-0162OC, 10.1016/j.ejpb.2018.10.020, 10.1016/j.celrep.2019.12.015, 10.3389/fimmu.2020.00836, 10.3389/fimmu.2020.01609, 10.3390/ijms22062938, 10.3389/fimmu.2021.657261, 10.3389/fimmu.2021.663695, 10.3390/ijms222212189). PBMCs were isolated and seeded at 2.5 × 106 cells/mL in Roswell Park Memorial Institute (RPMI) 1640 medium and supplemented with 10% AB-human serum and plated onto non-treated tissue culture plates. 10% AB-human serum promotes the differentiation of primary human monocytes into primary human macrophages (hMDMs). To obtain pure hMDMs, the cells were differentiated and cultured over 7–10 days at 37°C and 5% CO2 prior to experimentation. Non-adherent cells were removed by washing the cells every 2–3 days (non-adherent monocytes and other cells washed away whereas macrophages adhered to the culture plates within the first 24-72 hours). The purity of the hMDMs was routinely >95%, as assessed by flow cytometry. We have now added extra details into the current manuscript (lines 692-696). 

Reviewer 2 Comment 4: ‘The protocol of ROS measurement is not clear, what reagent is used’?

Author’s Response to Reviewer 2 Comment 4: Regarding the method to measure ROS production, the ‘Methods’ section was clarified to reflect the suggested changes. Specifically, the reagent used for ROS measurement, DHR123, has now been explicitly detailed to enhance the clarity of our protocol (line 797-818).

Reviewer 3 Report

Comments and Suggestions for Authors

Dear Sir/Madam,

I am writing to thank you very much for providing with the opportunity to review the ijms-2810327 manuscript. This is a very nice and interesting piece of work that attempts to explore the dynamic interactions between alveolar macrophages and infiltrating neutrophils during pulmonary infection by Mycobacterium tuberculosis. Although the study is oriented around M. tuberculosis, its findings could be rationally expanded to simulate multiple lung microbial challenges by diverse foreign species of interest.

The study describes a novel and sophisticated experimental platform that can be customized to dissect the dynamic inflammatory processes that microbialy challenged macrophages induce to surveilling neutrophils. The activated neutrophils exhibit a primed secretome profile, reduced migratory potential and upregulated bioenergetic markers, all associated with a primed inflammatory response.

The findings presented are based on an array of diverse experimental approaches and the study includes several important layers of findings, including the potential roles of IL-8 and CXCR2 as regulators of the neutophil local responses.

Although the study does not present a specific mechanistic finding and does not explore any of the specific soluble mediators identified as potential inducers neutrophil inflammation, the novelty of its customizable platform sheds new light to several molecules that could be potential pharmacological inflammatory targets for advanced biological drugs towards the therapy of tuberculosis and other lung infectious agents.

I think that this study is worth getting published with a minor revision.

i) You may wish to consider including TB in your manuscript title:

e.g. Human macrophages induce an inflammatory bystander neutrophil metabolism and effector response upon TB challenge

ii) The fonts in all Figs. should be increased by 1-2 points, i think that in the existing .pdf format at 100% zoom (real dimensions), the current ones are quite small even for high definition/retina monitors.

Author Response

Thank you very much for your thorough review of our manuscript. We are very pleased that you found the research valuable and informative. We have now reviewed your comments and have responded accordingly below.

Reviewer 3 Comment 1: ‘You may wish to consider including TB in your manuscript title’.

Author’s Response to Reviewer 3 Comment 1: Thanks for this recommendation. We have now amended the title to reflect this.

Reviewer 3 Comment 2: ‘The fonts in all Figs. should be increased by 1-2 points, i think that in the existing .pdf format at 100% zoom (real dimensions), the current ones are quite small even for high definition/retina monitors’.

Author’s Response to Reviewer 3 Comment 2: Thanks for this suggestion. We have now increased the font on the x-axis and the y-axis for all data graphs. We have included these new figures for resubmission.

Reviewer 4 Report

Comments and Suggestions for Authors

The paper by Murphy et al. is about the effect of human macrophages on neutrophil function during lung infections. The authors show that macrophages stimulated with Mycobacterium tuberculosis or LPS secrete cytokines and chemokines that activate neutrophils, enhance their metabolism, reduce their migration, increase their ROS production, NET formation, and cytokine secretion. The authors suggest that this macrophage-neutrophil axis may play a role in the pathological inflammation and tissue damage observed in severe pneumonia and other lung diseases. The paper also explores the immunometabolic profiles of neutrophils in response to different stimuli and pathogens.

I have some concerns below:

1. The paper used an in vitro model of human macrophages and neutrophils, which may not fully reflect the complex interactions and dynamics of these cells in vivo during lung infections.

2. The paper examined the effect of macrophage conditioned medium on neutrophil function after one hour of incubation, which may not capture the long-term effects of macrophage-neutrophil communication or the changes in neutrophil phenotype and function over time.

3. The paper could not completely remove the residual LPS from the LPS-stimulated macrophage conditioned medium, which may have a direct effect on neutrophil activation and migration, independent of the macrophage-derived factors.

4. The paper measured the levels of 18 cytokines and chemokines in the macrophage conditioned medium, but there may be other soluble mediators that are involved in modulating neutrophil biology that were not assessed. Some Seq technology is needed.

Author Response

Thank you very much for your thorough review of our manuscript. We are very pleased that you found the research valuable and informative. We have now reviewed your comments and have responded accordingly below.

Reviewer 4 Comment 1: ‘The paper used an in vitro model of human macrophages and neutrophils, which may not fully reflect the complex interactions and dynamics of these cells in vivo during lung infections’.

Author’s Response to Reviewer 4 Comment 1: We completely agree. Assessing these complex in vivo interactions in in vitro models is a very challenging task. This was also very evident when we conducted and published a literature review on the topic in late 2022 (DOI: 10.3389/fimmu.2022.984293). In this review, we discussed known interactions between neutrophils and other immune cells in an infectious disease setting and speculated what it could mean during active TB disease. It was very evident during this process that further research examining the link(s) between neutrophils and other immune cells was warranted in the field. We subsequently set out to fill this research gap first by specifically examining how human macrophages (immune cells that our laboratory specialises in the most) affect human neutrophil function. Our aim in the future is to examine how various immune cells affect neutrophil function in the human lung in the hope that we can learn more about the complex interactions and dynamics of these cells in vivo during lung infections.

Reviewer 4 Comment 2: ‘The paper examined the effect of macrophage conditioned medium on neutrophil function after one hour of incubation, which may not capture the long-term effects of macrophage-neutrophil communication or the changes in neutrophil phenotype and function over time’.

Author’s Response to Reviewer 4 Comment 2: Although we concur with the reviewer, we specifically set out to establish a model to study the early effects of macrophage and neutrophil interactions in an infectious disease setting. A publication (doi: 10.1016/j.cmet.2021.03.018) comparing neutrophil function and metabolism in acute and chronic inflammation also supported our early hypothesis that the priming of metabolic function by the inflammatory environment, underpins rapid neutrophil responses. They show that neutrophils undergo gluconeogenesis and glycogenesis in response to acute physiological stress and activation, and metabolic shifts that are required for fast energy production, function and survival. Our study further highlights and emphasizes the potency with which MtbCoM or LPSCoM can activate neutrophils and shows that an initial stimulatory or activation event could potentially lead to a rapid downstream cascade of activated neutrophils. We have detailed this study in the discussion section (lines 503-549).The reviewer is correct however and examining the long-term effects of the conditioned medium on neutrophil function would also be very informative for future studies. In support of this, and in the study just discussed, the authors also show that chronic inflammation in their neutrophil model dysregulates various cellular processes which have profound inhibitory effects on neutrophil function and survival. We have also mentioned this in the discussion section (lines 539-549).

Reviewer 4 Comment 3: ‘The paper could not completely remove the residual LPS from the LPS-stimulated macrophage conditioned medium, which may have a direct effect on neutrophil activation and migration, independent of the macrophage-derived factors’.

Author’s Response to Reviewer 4 Comment 3: We agree that residual LPS could direct effect on neutrophil activation and migration. It is extremely difficult to remove residual LPS from conditioned medium without introducing other confounding factors due to the treatment. We have addressed this shortcoming to the reader that we cannot rule out a direct effect of residual LPS on the neutrophils in our model; see the ‘Study Limitations’ section (line 629). Luckily however, the conditioned medium from macrophages stimulated with Mtb was processed further with this issue in mind. First, the conditioned medium was centrifugated (to pellet any bacteria) and the supernatants were subsequently filtered to eliminate spill over of Mtb directly affecting neutrophils. The details of this can be found in the methods section (lines 697-717).

Reviewer 4 Comment 4: ’The paper measured the levels of 18 cytokines and chemokines in the macrophage conditioned medium, but there may be other soluble mediators that are involved in modulating neutrophil biology that were not assessed. Some Seq technology is needed’.

Author’s Response to Reviewer 4 Comment 4: We agree that RNA sequencing should be the next pathway taken for the current research. From the outset, the primary focus of the current paper was to elucidate the effect of macrophage conditioned medium on neutrophil function at the protein and cellular level, which we strictly adhered to. We concur that an RNA sequencing dataset would be very informative for the field. Although this was outside the scope of the current study and budget, there are already several large sequencing datasets available for human neutrophils in discreet disease settings. In fact, we are currently working on data repurposing from existing sequencing to examine the transcriptome and interactome between neutrophils and other myeloid derived cells in the lung.

Reviewer 5 Report

Comments and Suggestions for Authors

This is interesting and important work on the macrophage-neutrophil axis. I only have minor comments:

Due to the large number of abbreviations, I propose to compile a list of abbreviations used in this work.

Abstract; line 20 I suggest using "unclear" or "only partially known" instead of "unknown" there are several reports on this topic

Introduction; Please add a short justification for using LPS as a second stimulator, maybe from lines 215-216?

Discussion; Line 493; not always; please find some reports on neutrophil lifespan;

Blood. 2010;116(4):625-627), Biomedycyna 2020, 8, 278; doi:10.3390/biomedicines8080278, doi: 10.4049/jimmunol.1102863

Author Response

Thank you very much for your thorough review of our manuscript. We are very pleased that you found the research valuable and informative. We have now reviewed your comments and have responded accordingly below.

Reviewer 5 Comment 1: ‘Due to the large number of abbreviations, I propose to compile a list of abbreviations used in this work’.

Author’s Response to Reviewer 5 Comment 1: Thank you for your suggestion. A list of abbreviations has now been attached in the ‘Supplementary information’ section. 

Reviewer 5 Comment 2: ‘Abstract; line 20 I suggest using "unclear" or "only partially known" instead of "unknown" there are several reports on this topic’.

Author’s Response to Reviewer 5 Comment 2: Thank you for your suggestion. ‘unknown’ has now been amended to ‘unclear’ in line 21 as per your recommendation. 

Reviewer 5 Comment 3: ‘Introduction; Please add a short justification for using LPS as a second stimulator, maybe from lines 215-216?’

Author’s Response to Reviewer 5 Comment 3: We have carefully revised the results section 2.1, incorporating a concise rationale for the inclusion of LPS, as suggested, lines 137-140. 

Reviewer 5 Comment 4: 'Discussion; Line 493; not always; please find some reports on neutrophil lifespan;'

Author’s Response to Reviewer 5 Comment 4: Thank you for your valuable suggestion. We recognise that the duration of neutrophil lifespan in vivo may vary, as studies have reported disparate findings (Blood. 2010;116(4):625-627), Biomedycyna 2020, 8, 278; doi:10.3390/biomedicines8080278, doi: 10.4049/jimmunol.1102863). In response to your comment, we have amended the suggestion that lifespan is less than 1 day and incorporated references to the mentioned reports on neutrophil lifespan in the Discussion section, line 635.